https://doi.org/10.1038/s41467-022-30335-2　　**OPEN**

# Diamond mirrors for high-power continuous-wave lasers

Haig A. Atikian[1], Neil Sinclair[1,2], Pawel Latawiec[1], Xiao Xiong [1,3], Srujan Meesala[1], Scarlett Gauthier[1], Daniel Wintz[1], Joseph Randi[4], David Bernot[4], Sage DeFrances[4], Jeffrey Thomas[4], Michael Roman[5], Sean Durrant[5], Federico Capasso [1] & Marko Lončar [1✉]

High-power continuous-wave (CW) lasers are used in a variety of areas including industry, medicine, communications, and defense. Yet, conventional optics, which are based on multilayer coatings, are damaged when illuminated by high-power CW laser light, primarily due to thermal loading. This hampers the effectiveness, restricts the scope and utility, and raises the cost and complexity of high-power CW laser applications. Here we demonstrate monolithic and highly reflective mirrors that operate under high-power CW laser irradiation without damage. In contrast to conventional mirrors, ours are realized by etching nanostructures into the surface of single-crystal diamond, a material with exceptional optical and thermal properties. We measure reflectivities of greater than 98% and demonstrate damage-free operation using 10 kW of CW laser light at 1070 nm, focused to a spot of 750 μm diameter. In contrast, we observe damage to a conventional dielectric mirror when illuminated by the same beam. Our results initiate a new category of optics that operate under extreme conditions, which has potential to improve or create new applications of high-power lasers.

[1] John A. Paulson School of Engineering and Applied Sciences, Harvard University, Cambridge, MA 01238, USA. [2] Division of Physics, Mathematics and Astronomy, and Alliance for Quantum Technologies (AQT), California Institute of Technology, Pasadena, CA 91125, USA. [3] Key Laboratory of Quantum Information and Synergetic Innovation Center of Quantum Information and Quantum Physics, University of Science and Technology of China, Hefei, Anhui 230026, China. [4] Pennsylvania State University Applied Research Laboratory, Electro-Optics Center, Freeport, PA 16229, USA. [5] Laser Technology and Analysis Branch, Naval Surface Warfare Center, Dahlgren Division, Dahlgren, VA 22448, USA. ✉email: loncar@seas.harvard.edu

High-power CW lasers are used in cutting, welding, and cleaning in construction and manufacturing[1–5], directed energy in military applications[2,6,7], medical surgery[2,8–11], communications[12–14] and sensing[15,16], ignition[17,18], mining[19–21], as well as atomic-molecular-optical physics and spectroscopy[2,22–25], among others. These applications require optical components, in particular mirrors, that withstand high CW or quasi-CW optical powers for directing light from the laser to the target. Conventional dielectric mirrors use multi-layered coatings[26] or nanostructured thin films[27] to engineer their reflection spectrum. The former utilizes alternating thin-film layers of varying refractive index and thickness to generate an interference effect at a desired wavelength and polarization, while the latter leverages localized or guided resonances to achieve high reflectivity. Yet, imperfections and defects in, or interfaces between, thin films form sites where laser energy can be absorbed[28–31]. Using high-power CW laser light, absorption at these sites generates significant heat, causing melting or thermal stress between film layers. This thermal loading degrades the optical performance and produces irreversible damage to the mirror. We overcome this limitation of multilayered, multi-material, optical coatings by surface-engineering the optical response of single-crystal diamond to demonstrate it as a highly reflective mirror for high-power CW lasers. Diamond is utilized due to its exceptional properties: relatively high refractive index (2.4), wide bandgap (5.5 eV), high mechanical hardness and chemical resistance, and the highest material thermal conductivity at room temperature (2200 W/K·m)[32–34]. Consequently, diamond materials, in particular optics, can be used in a diverse range of applications and operating environments, see e.g., Refs. [35–39] and references therein. Photonic crystals and metamaterials have emerged as a promising technology for tailoring properties of optical beams[40–44]. These are typically composed of two-dimensional arrays of holes or rods in a thin-film layer that allow engineering the spatial distribution of amplitude, phase, and polarization response of an optical element[45–48]. Many optical components have been realized using this approach, such as mirrors, lenses, and polarization optics[49–54]. Conventionally, planar photonic crystals and metamaterials are formed by nanopatterning a high-index dielectric (or metallic) film that has been deposited on a low-index substrate to leverage the index contrast needed to support optical resonances[55,56]. Yet, these suffer from the same power handling limitations as conventional multilayer thin films. We avoid this by creating nanostructured mirrors from a monolithic substrate, strategically one with exceptional properties, creating a mirror that withstands high-power CW laser light.

## Results

As illustrated in Fig. 1a, a mirror is formed by a planar lattice of identical "golf tee"-shaped columns that are etched into a diamond surface. For comparison, Fig. 1b depicts a traditional multilayered optical coating on a substrate. It is possible to control the properties of the mirror by engineering the geometry of each column in the lattice. Referring to Fig. 1c, this involves varying the angle $\alpha$ of the top region, radii $r_{disc}$, $r_{min}$, $r_{support}$, column height $h$, and the pitch, i.e. the center-to-center distance between the columns. The high reflectivity of the structure is attributed to a lattice resonance that is dominated by lateral leaky Bloch modes[57]. These guided resonances are confined to the top region of each column, as seen in Fig. 1e, and are not supported by the narrowest part of the column which facilitates mode confinement. To achieve reflection, parameters of the periodic array must satisfy the well-known grating equation $d\left(\sin\theta_i - \sin\theta_m\right) = m\lambda$, where $d$ is the grating period or pitch, m is an integer representing the diffraction order, the angles of the incident beam and the $m^{th}$ diffracted order are $\theta_i$ and $\theta_m$, while $\lambda$

is the wavelength of the incident beam[58]. The first diffraction orders are coupled into the resonance supported by the top portion of the column, and then out-couple to the zeroth order of the grating in both reflected and transmitted direction. With proper design of the columns, incident angle and wavelength of light, the transmitted beams will interfere destructively resulting in perfect reflection, as indicated by the uniform phase front above the mirror in Fig. 1e.

Intuitively, the lateral mode guiding can be understood in the following way. Each column comprises three distinct regions of effective refractive index, labeled $n_1$, $n_2$, and $n_3$ in Fig. 1c. The shaded red region labeled $n_2$ contains the top region of the column. It effectively acts as the high-index layer since it contains more diamond per volume than the other regions and serves as the guiding layer for supporting leaky Bloch mode resonances in the device. The shaded yellow region labeled $n_3$ contains the narrow parts of the column, thereby acts as a lower-index layer ($n_3 < n_2$), and will provide optical confinement for the guiding layer $n_2$. The shaded yellow region above labeled $n_1$ is air, providing the condition that the effective indices of each region $n_2 > n_1$, $n_3$ to support guided optical resonances[59,60].

Simulations using a finite-difference time-domain (FDTD) solver, are performed to optimize the structure for maximum reflection at normal incidence. Figure 1d shows a simulated diamond mirror reflection spectrum for varying design angles α. We target an operating wavelength of 1064 and 1070 nm, which are technologically relevant for high power lasers, while dimensions of the columns are chosen to give the widest bandwidth of high reflectivity around the target wavelength, see Fig. 1d. More details on the FDTD simulations and reflection spectra for other relevant dimensions are described in Supplementary Discussion 1.

To realize these complex 3-D structures across a wide area, we use an unconventional, yet scalable, angled-etching nanofabrication technique for single-crystal chemical vapor deposition diamond, as illustrated in Fig. 2 and described in its caption. Concisely, we utilize oxygen-based reactive-ion beam angled etching (RIBAE)[61]. Complete fabrication details are discussed in the Methods.

An optical image of a fabricated mirror is shown in Fig. 2b. A scanning electron microscope (SEM) image of the mirror is presented in Fig. 2c, with a zoomed image displayed in Fig. 2d. Both images show the "golf tee"-shaped columns in a uniformly spaced array. The area of the diamond mirror is 3 mm × 3 mm, with nearly identical device geometry from one edge of the mirror to the other. The ability to precisely, and uniformly, fabricate nanometer-scale geometries across a large surface is enabled by the RIBAE technique.

The reflection spectrum of a diamond mirror is measured using a procedure outlined in the Methods. The result is shown in Fig. 3a, showing excellent agreement with the predictions of the FDTD simulations for $\alpha = 70°$ and the rest of our target design parameters, see the caption of Fig. 1d. An absolute reflectivity of 98.9 ± 0.3% at 1064 nm is measured, with uncertainty owing to the accuracy of the optical power detector. Approximately 0.5% of the optical power is transmitted through the backside-polished diamond substrate, while the remaining 0.6% is loss, likely due to scatter rather than absorption. Measurements of high-quality factor resonators produced in diamond using RIBAE have suggested little surface absorption[61–63]. Moreover, a reflectivity of greater than 98% is observed across a 10 nm bandwidth around 1064 nm, also consistent with simulations.

Beam profile measurements are performed on the reflection from the diamond mirror to ensure an incident 1064 nm-wavelength Gaussian beam is not distorted. See the Methods for more details. A 2-D plot of the power distribution of the reflected beam is presented in Fig. 3b, with cross-sectional profiles for each

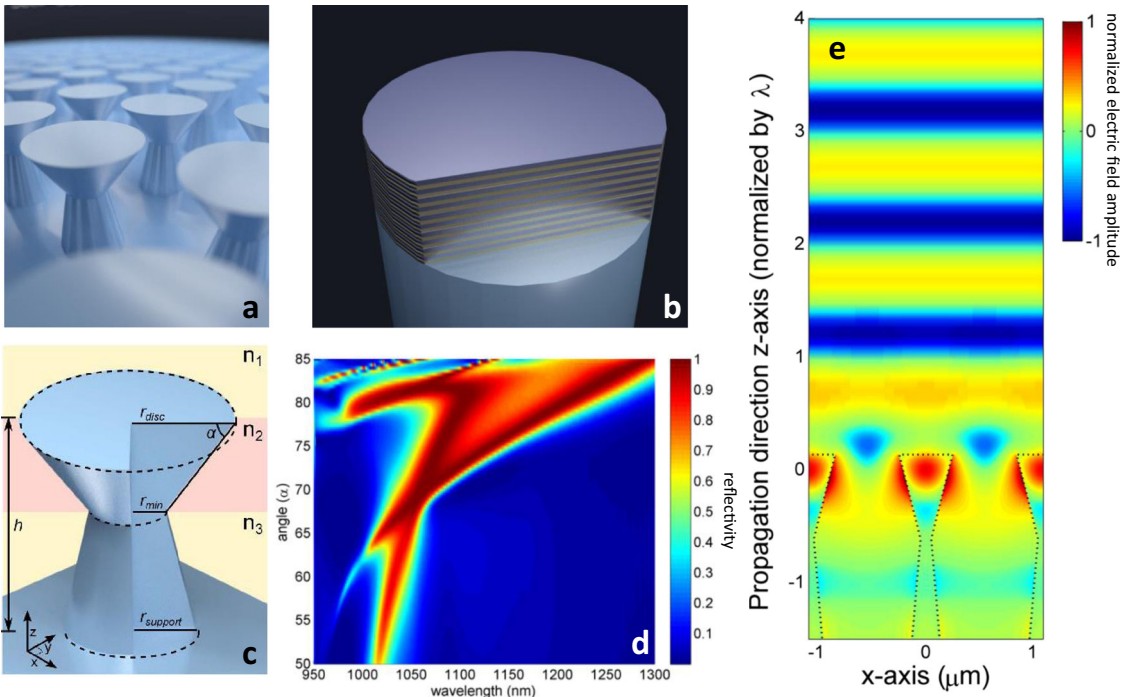

**Fig. 1 Design and simulation of a mirror in single-crystal diamond. a** Graphical depiction of a diamond mirror with the "golf tee"-shaped columns arranged in a hexagonal lattice. **b** Typical multilayered optical coating deposited onto a substrate. **c** Schematic of the "golf tee" columns that comprise the diamond mirror, with all relevant dimensions labeled: angle $\alpha$, radii $r_{disc}$, $r_{min}$, $r_{support}$, and total height $h$. The shaded yellow region labeled $n_1$ is of lowest refractive index (air), the red region $n_2$ contains the top portion of the column that features optical resonances and is of highest refractive index, while the yellow region $n_3$ is of lower refractive index and supports the top portion of the column. **d** Diamond mirror reflection spectrum at normal incidence for varying design angles $\alpha$, with $r_{disc} = 250$ nm, $r_{min} = 50$ nm, $r_{support} = 250$ nm, pitch 1.1 µm, and $h = 3$ µm. Colors indicate reflectivity. **e** Standing-wave pattern illustrating the reflected wavefront from a diamond mirror at a wavelength of 1064 nm. Mode is confined in the top portion of the columns due to lattice resonance. Colors indicate the electric field amplitude. Photo credit for panels (**a**) and (**b**): P. Latawiec, Harvard.

axis overlaid with an independent Gaussian fit. These measurements are used to determine the 3-D power distribution of the reflected beam, as depicted in the inset of Fig. 3b, further illustrating the absence of any beam distortion. Note that our FDTD simulations show that the nanostructured diamond surface maintains a uniform phase-front for reflected beams, see Supplementary Fig. 6.

Next, we design and fabricate a 3 mm × 3 mm mirror with measured reflectivity of 96% at 1070 nm. The mirror is mounted to a water-cooled stage at 18 °C and irradiated for 30 s using CW laser light with varied power to assess its laser-induced damage threshold (LIDT) at this wavelength. Note that typical optics use beam expanders to mitigate laser damage, while here we focus the beam to a 750 µm ($1/e^2$) diameter, corresponding to hundreds of periods of the "golf-tee" lattice. This spot size represents a reasonable beam diameter that would be used in a practical laser system, corresponding to a Rayleigh length of roughly 1.66 m for a Gaussian beam at this wavelength. Smaller spot sizes can be used to increase the power density; however, we attempt to perform the LIDT tests with beam waists, and associated thermal loading, that would be present in a realistic optical system. Moreover, we perform tests at normal incidence, ensuring maximum energy is directed at the mirror. Optical and thermal images taken during the LIDT tests are shown in Fig. 4a–e, while all details of the testing mirror, setup, and procedure are provided in the figure caption and Methods. Thermal and optical videos of the tests are shown in Supplementary Videos 1–6. The hot spot in the images suggests that the laser power which is not reflected is, rather, transmitted through the mirror, heating the underlying water-cooled stage. Optical microscope and SEM imaging after the tests indicate no damage or change in surface morphology. A

wide-angle view of the diamond mirror after LIDT testing is shown in Fig. 4f. Moreover, we measured the mirror reflectivity after the damage tests, finding that it is also maintained. Therefore, we cannot determine the LIDT for the diamond mirror using up to 10 kW of CW laser light, demonstrating its robustness for high power applications.

To put our result into context, we repeat the LIDT tests using a standard dielectric mirror of 99.5% reflectivity. Optical and thermal images are shown in Fig. 4g–k, with further details of the testing, dielectric mirror properties, and setup described in the figure caption and Methods, while a thermal video of the test at 10 kW power is shown in Supplementary Video 7. As the power is increased, the hot spot rapidly increases in temperature due to absorption, poor thermal conductivity, and expansion of the dielectric coatings[30], leading to damage under 10 kW of irradiation. This is confirmed by an optical image of the mirror taken after the tests, shown in Fig. 4l, which suggests inferior performance to the diamond mirror for optics irradiated with high-power CW laser light.

## Discussion

We demonstrated highly reflective monolithic diamond mirrors that withstand high-power CW laser light. Our results are supported by beam profile measurements and numerical modeling in which no distortions in the reflected laser beam were inferred. Damage testing demonstrated the ability of a diamond mirror to operate under 10 kW CW laser illumination, contrary to that using a standard dielectric mirror, which cannot survive the high thermal loading at these powers. The damage owes to the high power of the CW beam across a spot of 750 µm diameter, differing from tests using high-peak-power femtosecond- and

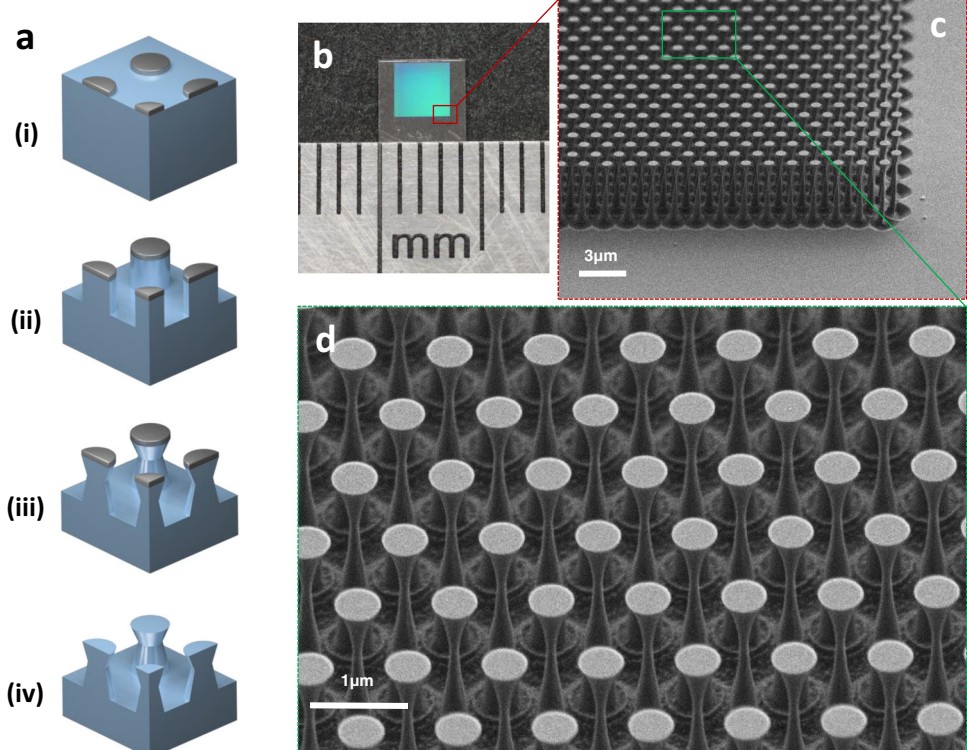

**Fig. 2 Diamond surface fabrication and images. a** Schematic of the reactive ion beam angled etching (RIBAE) fabrication process. (i) Etch mask is patterned onto the diamond sample surface. (ii) Top-down etch with the sample mounted perpendicular to the ion beam path on a rotating sample stage. (iii) Sample is tilted during etching to obtain the target angle α with respect to the direction of the ion beam, uniformly etching underneath the etch mask. (iv) Mask removal yields an array of 3-D nanostructures etched into the surface of diamond. **b** Optical image of the diamond mirror on a 4.2 mm × 4.2 mm diamond crystal. Each division on the ruler is 1 mm. Photo Credit: H. A. Atikian, Harvard. **c** SEM image of the diamond mirror taken at 60° from normal. **d** Zoomed SEM image of the mirror taken at 40° from normal.

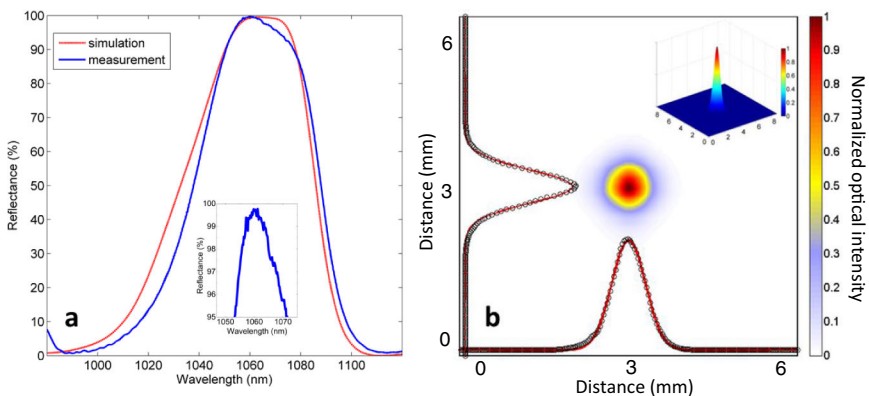

**Fig. 3 Optical characterization of a diamond mirror. a** Reflection spectrum of a diamond mirror, blue line is measurement data and red line is FDTD simulation. Absolute reflectivity of 98.9 ± 0.3% is measured at 1064 nm using a DBR laser. Inset shows a zoom-in of the measured spectrum around its maximum. **b** Beam profile measurement taken of the reflection from the diamond mirror using a scanning-slit profiler. Axes show cross sections of the reflected beam (black circles) with overlaid Gaussian fit (red). Fit yields a 4σ beam width of ~1.5 mm. Distance refers to the amount traveled by the slit relative to its initial position. Inset shows a 3-D perspective of the reflected beam, with axes (and its units) identical to the main figure. Colors indicate normalized optical intensity.

picosecond-duration pulses from mode locked lasers, which cause damage (to dielectrics, including diamond) predominantly due to impact ionization and dielectric breakdown. Further tests of the damage thresholds of the monolithic diamond mirror against custom high-power mirrors, e.g. those relying on ion-beam-sputtered coatings on varied dielectrics, including diamond, would be valuable to determine the full extent of our mirror relative to the state of the art, see e.g., Refs. [64–66]. Future work involves extending our approach to optical components for CW high-power lasers at other wavelengths, which could benefit several applications[2–4,6–8]. Finally, we emphasize that our mirror technology is not limited to diamond alone, as reflectors can be fabricated from a wide variety of materials. For example, monolithic mirrors leveraging the extremely large bandgap (~9 eV) of fused silica could benefit ultrafast laser applications.

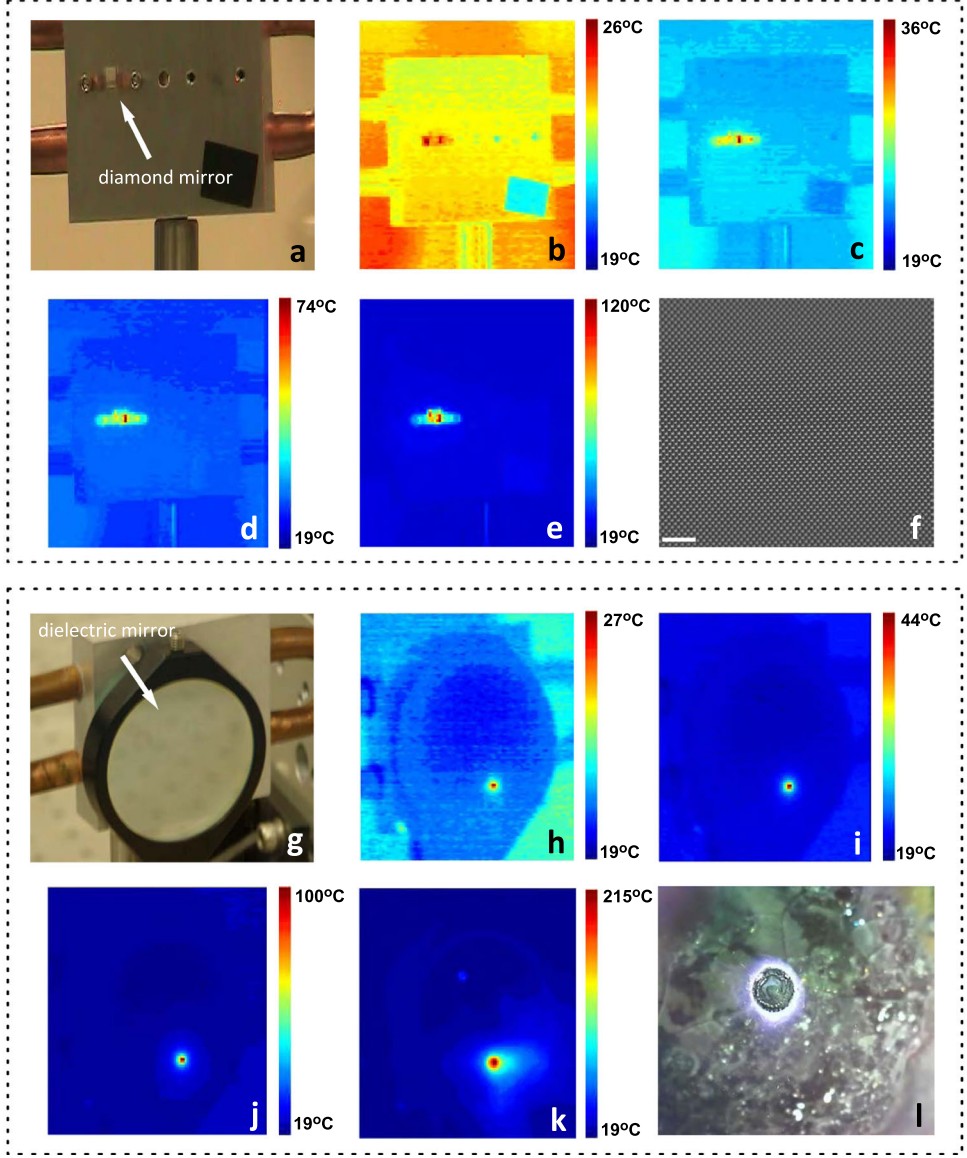

**Fig. 4 Laser-induced damage testing of diamond and dielectric mirrors. a** Optical image of a diamond mirror mounted on a water-cooled stage that was taken prior to testing. **b**–**e** Thermal images of the diamond mirror irradiated by 0.5, 2.5, 5, and 10 kW, respectively, of continuous-wave laser power. Color bar shows the temperature of the setup with varying scale for each image. Temperature accuracy is ±2°. Hot spot corresponds to the position of the beam (on the diamond mirror). At increased power levels a small fraction of optical power leaking through the backside of the diamond mirror results in heating of the stage. **f** Wide-area SEM image of the diamond mirror shows no damage after testing. Scale bar is 5 μm. **g** Optical image of a corresponding dielectric mirror mounted on the water-cooled stage. **h**–**k** Thermal images of the dielectric mirror irradiated by 0.5, 2, 6, and 10 kW, respectively, of CW laser power. Damage ensues at 10 kW of power due to thermal stress. **l** Image of damage region of dielectric mirror taken after testing shows a several mm-sized hole where the laser beam ablated the dielectric. Photo Credit for panels (**a**), (**g**) and (**l**): S. DeFrances, Penn State EOC.

## Methods

**Diamond mirror fabrication**. To realize the complex 3-D column structures across a wide area, we employ an unconventional, yet novel and scalable, angled-etching nanofabrication technique utilizing a reactive ion beam etching (RIBE) process. RIBE is a derivative of ion beam etching (IBE) in which a broad-area ion beam source is used to collimate and direct a beam of high energy ions from a gas source. The distinction of RIBE is that the plasma source is comprised of reactive gases, whereas IBE is limited to noble gases such as Ar, Xe, or Ne. We use $O_2$ as a reactive gas to etch diamond. Ions are extracted from the plasma source using a set of electrically biased grids typically made from Mo. Voltages applied to these grids, along with the plasma source, dictate the energy, flux, and divergence of the ions. Typically, the uniformity of the ion beam can be greater than 95% across the ion beam source diameter, and the size of the sample (e.g. wafer or crystal) to be processed is only restricted by the cross-section of the beam.

Figure 5 depicts the RIBAE etching procedure that is used to create a diamond mirror[61]. The process begins by pattering an etch mask onto the diamond surface followed by a top down etch with the sample mounted perpendicular to the ion beam

path on a rotating sample stage, see Fig. 5b(i). Once the desired etch depth has been reached, the sample is tilted to an angle α with respect to the path of the ion beam, and the diamond columns are uniformly undercut in all directions, see Fig. 5b(ii). The etch mask is then removed to reveal the final structure depicted in Fig. 5b(iii).

We now describe all steps of the fabrication process in detail. A mirror is created from a type IIa single-crystal diamond from Element 6, grown by chemical vapor deposition with less than 5 ppb nitrogen concentration. The diamond sample is cleaned in a boiling mixture of equal parts sulfuric, nitric, and perchloric acid[61,62]. The etch mask is constructed as follows. First, a 70 nm-thick layer of Nb is deposited onto the surface of the sample by DC magnetron sputtering, followed by spin coating with hydrogen silsesquioxane (HSQ) resist. An array of circles in a hexagonal grid is created in the HSQ by performing 125 keV electron beam lithography following and developed using a 25% tetramethylammonium hydroxide solution. Finally, a top-down etch of the Nb film is performed in a UNAXIS Shuttleline inductively coupled plasma reactive ion etcher (ICP-RIE) with the following parameters: 400 W ICP power, 250 W radio frequency (RF) power, 40 sccm Ar flow rate, 25 sccm $Cl_2$ flow rate and $8 \times 10^{-3}$ Torr process pressure.

The rest of the fabrication follows the RIBAE process using a Kaufman & Robinson 14 cm RF-ICP ion beam source. RIBAE parameters are: 200 V beam potential, 26 V accelerator potential, 85 mA beam current, ~155 W ICP power, 37 sccm $O_2$ flow rate, and $7.5 \times 10^{-4}$ Torr process pressure. A non-immersed electron source neutralizer is used to neutralize positive ions from the beam. The neutralizer is mounted on the side of the ion source with the emission current set to 1.25x of the ion source beam current, and with an Ar gas flow of 10 sccm.

Top-down RIBE of the sample is performed to achieve the desired depth of the structures followed by removal of the HSQ mask via hydrofluoric acid (HF). Nb does not react with HF, leaving the Nb mask intact to serve as the mask for the angled-etching process. The reasoning for the 70 nm-thick Nb mask is two-fold. First, it is an excellent etch mask for oxygen plasma, providing ample selectivity to create the desired structures without significant mask erosion. Second, a thin mask is required such that when the sample is tilted, the height of the resist should not shadow neighboring nanostructures. This restriction puts an ultimate limit to how close patterns can be relative to each other (i.e. this restricts the pitch) while still allowing undercut columns to be created.

RIBAE is performed at the design angle α (e.g. 70°) until the desired undercut is achieved and the target column dimensions are realized. This is followed by the removal of the Nb mask using buffered chemical polish (BCP) which consists of two parts 85% phosphoric acid to one part 49% hydrofluoric acid to one part 70% nitric acid. The sample is then rinsed in deionized water, followed by a solvent clean with acetone and isopropyl alcohol. The key characteristic of this technique is the remarkable uniformity across a wide area, potentially as large as 200 mm in diameter, limited only by the size of the ion beam source used.

### Experimental setup for reflection spectrum and beam profile measurements.
The reflection spectrum of a diamond mirror is measured using the setup outlined in Fig. 6. Broadband light is generated using a 1065 nm superluminescence diode (SLD, InPhenix IPSDD1004C), is collimated and directed, using broadband silver mirrors (Thorlabs PF10-03-P01), to a 50:50 beamspiltter (Thorlabs CM1-

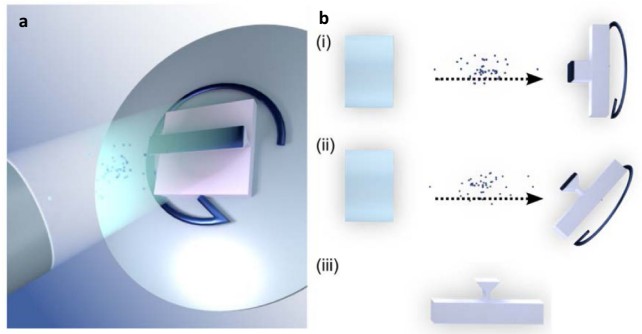

**Fig. 5 Reactive ion beam etching (RIBAE). a** Graphical depiction of RIBAE. **b** RIBAE fabrication steps (i) Top-down etching of a diamond sample mounted perpendicular to the ion beam path on a rotating sample stage. (ii) Sample is tilted to obtain an acute angle between the sample and ion beam, uniformly etching underneath the etch mask. (iii) Mask removal yields undercut nanostructures from a bulk substrate.

BP145B3) after passing through a lens (Thorlabs AC254-300-C-ML) which focuses the beam at the diamond mirror (or reference mirror, see below). Reflected light is directed by the beamsplitter to an identical lens that collimates the beam and directs it to an optical spectrum analyzer (Yokogawa AQ6370). We use a lens of long (300 mm) focal length to ensure the diameter of the beam at the diamond mirror is less than 1 mm, which is much smaller than the patterned area on the diamond crystal (3 mm × 3 mm). After a spectrum of a broadband silver mirror (Thorlabs PF10-03-P01) is measured for reference, the diamond mirror is measured, and its spectrum is determined with normalization to the reference.

A more precise measurement of reflectivity at 1064 nm is performed by replacing the SLD with a 1064 nm distributed Bragg reflector (DBR) laser of 10 MHz linewidth (Thorlabs DBR1064S) and the optical spectrum analyzer with a free-space optical photodetector (Newport 918D-SL-OD3R sensor attached to a Newport 1936R power meter, averaging mode). A reference measurement is taken using a Nd:YAG laser and a mirror of 99.8% reflectivity (Thorlabs NB1-K14) to precisely determine the reflectivity of the diamond mirror at 1064 nm.

Beam profile measurements are performed using the 1064 nm DBR laser with a scanning slit beam profiler (Thorlabs BP209-IR) replacing the photodetector. A least-squares method is used to fit ($\chi^2 = 0.002$) the x-y Gaussian cross-section profile of the beam, see Fig. 3b.

### Setup and details of laser damage testing experiments.
The laser-induced damage threshold (LIDT) tests of the diamond and dielectric mirrors are assessed at the Pennsylvania State University Applied Research Laboratory, Electro-Optics Center. The testing is performed using a 1070 nm multimode fiber laser from IPG Photonics, capable of providing up to 10 kW of continuous wave laser light. The diamond mirror is designed and fabricated to reflect light of 1070 nm wavelength and is clamped to a water-cooled aluminum stage (Aavid 416401U00000G) using Cu clamps. The dielectric mirror (Thorlabs BB2-EO3) is also mechanically clamped to the water-cooled aluminum stage. The chiller used to cool the stage is at a temperature of 18 °C and flows at approximately 7.5 liters per minute. A lens of 500 mm focal length focuses the laser to a spot of 750 μm ($1/e^2$) diameter at the diamond and dielectric mirrors. Inspection and digital image capture of the mirrors are performed during the LIDT tests using an off-axis optical camera. A Mikron M7600 thermal imaging camera (accuracy of ±2°, emissivity set to 0.97) is also used to monitor the temperature of the mirrors and aluminum stage throughout testing. It is directed 2–3° off normal incidence to avoid reflections returning into the laser.

The reflection spectrum of the diamond mirror is simulated and measured beforehand using the superluminescent diode and accompanying setup as described in the previous section, with results shown in Fig. 7, noting the reflectivity is 96% at 1070 nm. The spectrum of the 10 kW IPG laser is also shown in this figure (in arbitrary units), illustrating the overlap of the laser used during LIDT testing with the reflection spectrum of the diamond mirror under test.

The cross-section profile of the high-power beam from the IPG laser is measured using a Primes focus monitor, results are shown in Fig. 8 along with a Gaussian fit. The Primes focus monitor has a metal tip with a pinhole of 20 μm diameter that can be translated by a motorized stage to the desired location. The tip traverses the entire area of the beam, collecting a 2-D map of the beam profile.

The LIDT test is performed such that the mirrors are irradiated for 30 s with a constant laser power. The laser power is increased from 0.5 to 10 kW, and the test is repeated for each power level, see the caption of Fig. 4. During each LIDT test of the diamond and dielectric mirrors (i.e. for each power level), the temperature at the hot spot quickly reached a steady state once illumination began, and remained at that temperature until illumination ceased. That is, except for the 10 kW-test using the dielectric mirror, i.e. at the onset of damage of the dielectric mirror. The steady-state temperatures are shown in the thermal images of Fig. 4b–e and h–j. As

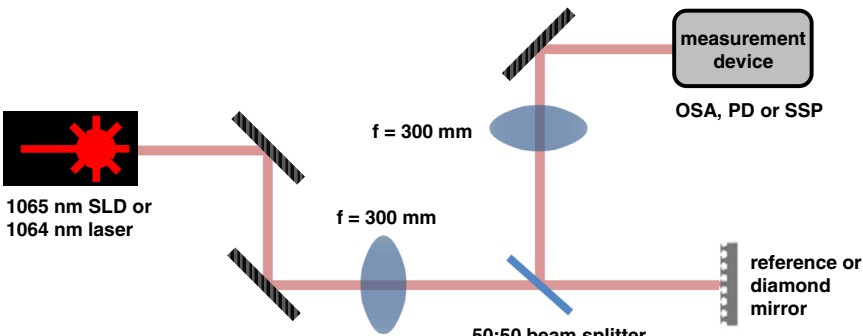

**Fig. 6 Schematic of the experimental setup used for measuring the reflection spectrum of a diamond mirror and beam profile measurements.** The reflection spectrum is measured using light from a 1065 nm SLD that is collimated and directed with broadband silver mirrors to a 50:50 beamsplitter after passing through a focusing lens. Reflected light from the diamond mirror, or a reference mirror, is directed to an optical spectrum analyzer (OSA) after passing through a defocusing lens. A 1064 nm DBR laser source and free-space optical photodetector (PD) replaced the diode and OSA for the more precise reflectivity measurements. The PD was replaced by a scanning slit profiler (SSP) for beam profile measurements.

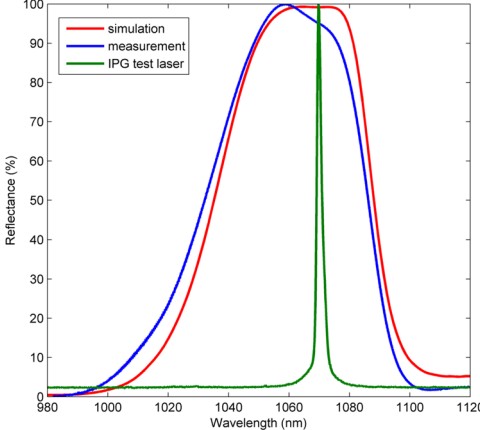

**Fig. 7 Reflection spectrum of the diamond mirror used for LIDT measurements at 1070 nm.** A diamond mirror measured (blue curve) and simulated (red curve) reflection spectrum at normal incidence. Green curve shows the spectrum of the 10 kW IPG laser used during damage testing plotted in arbitrary units.

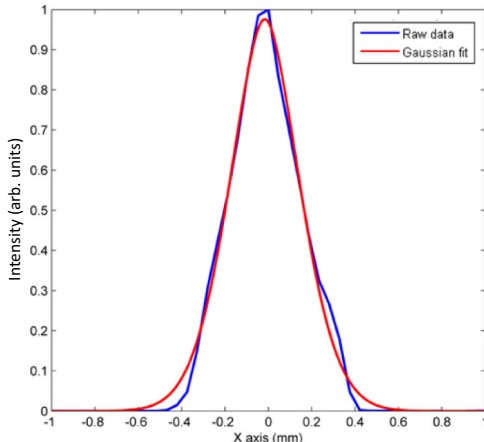

**Fig. 8 Beam profile of 1070 nm IPG LIDT test laser.** Beam profile is collected using a Primes focus monitor. The focus monitor has a metal tip with a 20 µm-diameter pinhole in the side. The rotating tip then traverses the entire area of the beam, collecting 2-D data of the beam profile. Blue line represents the raw data from the x-axis of the beam. The blue (red) line is the data (Gaussian fit).

illustrated in these figures, the hot spot temperature rose more quickly with increased laser power during the dielectric tests (see next paragraph). However, the temperature of the hot spot for the 10 kW tests of the dielectric mirror did not reach a steady state, but steadily increased over the duration of illumination until damage ensued, at which point the laser illumination was halted. The image in Fig. 4k is taken immediately after damage occurs.

As widely discussed in literature[29], the relatively low thermal conductivity of commonly used substrates and coatings for dielectric mirrors, led to the rapid temperature increase of the laser exposure site. Combined with the high, and varying, thermal expansion coefficient of these coatings, thermal stress resulted, and subsequent damage occurred. In contrast, tests of the single-crystal diamond mirror, which has high thermal conductivity, led only to heating of the aluminum plate, and no damage.

## Data availability

The datasets generated and analysed during the current study are available from the corresponding author on reasonable request.

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

## Acknowledgements

This work was performed in part at the Center for Nanoscale Systems (CNS), a member of the National Nanotechnology Coordinated Infrastructure Network (NNCI), which is supported by the National Science Foundation under NSF award no. 1541959. CNS is part of Harvard University. Laser-induced damage threshold of the diamond mirror was assessed at the Pennsylvania State University Applied Research Laboratory, Electro-Optics Center. This work was supported in part by the Air Force Office of Scientific Research (MURI, grant FA9550-14-1-0389), the Defense Advanced Research Projects Agency (DARPA, W31P4Q-15-1-0013), STC Center for Integrated Quantum Materials and NSF Grant No. DMR-1231319. N.S. further acknowledges support from the Natural Sciences and Engineering Research Council of Canada (NSERC), and the AQT Intelligent Quantum Networks and Technologies (INQNET) research program. P.L. was supported by the National Science Foundation Graduate Research Fellowship under Grant No. DGE1144152. The authors thank Daniel Twitchen and Matt Markham from Element Six for their support with the diamond samples, and Michael Haas for software assistance.

## Author contributions

H.A. and M.L. conceived the idea. H.A., X.X., and S.G. performed simulations. H.A. fabricated the mirrors. S.M. assisted with diamond preparation. H.A. and P.L. designed the experiment setup. H.A. performed optical characterizations. D.W. assisted with beam profile measurements. H.A. and N.S. analyzed and interpreted the data. J.R., D.B., S.D., J.T., M.R., and S.D. assisted with laser damage testing. H.A. and N.S. wrote the manuscript with the help of all co-authors. F.C. and M.L. supervised the project.

## Competing interests

H.A. and M.L are inventors on patent applications related to this work (U.S. No.: 10,727,072, date filed: May 2016, granted: Jul 2020) and (U.S. Application No.: 15/759,909, date filed: Sept 2016). The authors declare that they have no other competing interests.
