## [Peer Review File · Nature Communications]

REVIEWER COMMENTS

Reviewer #1 (Remarks to the Author):

The manuscript reports the design, fabrication and testing of a monolithic diamond high reflector that uses the resonant properties of an optimized nanostructure. Like moth eye anti-reflections structures, the aim is to make the optic from a single material, in this case one with extreme thermal and optical properties, to provide rapid heatsinking and also presumably to avoid differential expansion, thereby raising its power handling without damage. The work relies on the exquisite nanofabrication techniques of the team, which is world leading and has been used previously to develop a range of ground-breaking advances in diamond quantum optics, opto-mechanics and nanooptics. The mirror structures in this work are suitably backed up by a model of the em-field surface interaction. The main conclusions are that the fabricated mirror achieves a reflection coeff above 98% while withstanding 10 kW in 750 micron-diameter spot (equivalent to intensity 0.5 MW/cm² or linear damage threshold 0.13 MW/cm, to my calculations), a level of intensity that a sample off-the-shelf dielectric stack mirror was found to fail.

The results are noteworthy in that this is the first planar photonic crystal for high reflection (anti-reflection structures were demonstrated some years ago). On the point of the physics, the high reflectivity relies on a photonic resonance in the 'head' of the golf tee, which is an interesting outcome. The results may be of interest to those working in directed energy and materials processing applications. Since diamond transmits well in some areas of the infrared, including beyond 6 microns, the approach may be useful for long wavelength high power lasers such as CO₂ lasers. The authors suggest that it may also be helpful for CO₂ laser mirrors in the EUV lithography, although I believe that is a short-pulsed laser application so that more testing would have to be done to determine whether the mirrors are applicable to that application. A higher damage threshold is found compared to an off-the-shelf mirror Thorlabs broadband dielectric stack on glass, which hints (only) that the approach is promising for achieving higher damage thresholds than conventional mirrors. Most of the conclusions are well supported by the data – an exception is the aforementioned concern that the designs are not shown yet to be a benefit to the pulsed CO₂ laser application. The conclusion sentence "We expect our work to stimulate development of a new class of optical components for CW high-power laser applications across the optical spectrum." is perhaps also not yet justified by the data – see point #6 below.

The work is clearly explained (see also points below) and contains clear figures and useful supplementary material. I support publication in Nature Commun. with due consideration of the following points:

- 1) In the abstract, the sentence 'these applications are unsuitable for mode-locked lasers' is a little unclear – what applications are they referring to and is the intention to say that the applications are better served by CW lasers? This is a small point that could be addressed with some minor re-wording.
- 2) Line 88: the novel features of the etching method are not clearly specified – is it only the rotation of the substrate that is new, or is the plasma design novel too?
- 3) Fig 3a – it would be helpful to see a zoomed inset of the higher percentages, to make the 98% /

10nm claim more obvious.

4) The temperature images were taken with a FLIR camera. Please specify the model and state the uncertainty. Does the measurement rely on an emissivity value– if so, what value was used?

5) ‘The hot spot in the image suggests that 4% of laser power’ – this is confusing – does the hotspot in the video images arise from emission from the diamond or the cooling plate behind? How does the hotspot suggest ‘4%’? Annotation of the video point out the intersection of the beam with the diamond and the hotspot on the heatsink would be helpful.

6) The diamond mirror is compared to that for a broadband HR mirror – this is not a high bar for high power mirror performance. A better choice would have been a narrowband mirror, with their fewer layers and much higher damage thresholds (twice as high for Thorlabs and 0.1 MW/cm is standard for IBS coatings). It would be helpful if the authors had data available for a mirror that is designed for high power.

The study currently only shows that the microstructures mirrors work and provide reasonably high damage threshold. If an advantage over other coatings were to be demonstrated, a better experiment would be to compare their microstructured diamond mirror with a HR coating (eg., IBS) on a diamond substrate, in this way unambiguously showing the advantage of the structure over conventional coatings (the choice of substrate is also likely to play a role). I suggest the authors comment on this possibility in their discussion.

This is an interesting paper that points towards new possibilities for high power mirror structures. I look forward to seeing the final article in publication.

Rich Mildren

Reviewer #2 (Remarks to the Author):

The paper is well written and the described subject is original with respect to the state-of-the-art. The demonstrated properties of the designed and manufactured diamond mirror with ion-etched nanostructure offer an interesting opportunity for high-power CW-applications and are relevant to the field of high-power laser optics. I suggest to publish the paper after the authors have made a few minor changes to their manuscript. I have the following suggestions and comments for the authors:

- In Fig 1D, I suggest putting the quantity’s name “reflectivity” on the false color scale.
- In Fig 1E, I suggest putting the quantity’s name “electric field amplitude” on the false color scale.
- Line 82: “at vertical incidence” -> “at normal incidence”
- In Fig 3B, I suggest adding x- and y-axis and not just putting a scale bar into the diagram with the value only found in the captions. There is no label on the color bar. Also change “normalized optical power” to “normalized optical intensity”.
- Can the authors comment why they use single crystal diamond and not polycrystal diamond?
- The state-of-the-art regarding diamond as a high-power material for optical components is not described well. There are for example works that demonstrate the use of single crystal diamond lenses (with dielectric coatings) sustaining kW CW laser power: C. Holly, M. Traub, D. Hoffmann, C. Widmann, D. Brink, C. Nebel, T. Gotthardt, M. Ceyhun Sözbir, and C. Wenzel, “Monocrystalline CVD-

diamond optics for high-power laser applications," in High-Power Laser Materials Processing: Lasers, Beam Delivery, Diagnostics, and Applications V, Proc. SPIE, vol. 9741, 2016.

- Line 371: The "2" can be subscripted.
- Line 410: What is the goodness of the fit?
- Why is the laser focused onto the mirror for reflectivity measurements? This creates a spectrum of incidence angles around the normal incidence while with a collimated beam the angular spectrum is reduced.
- Extended Data Fig 4: Put label on vertical axis.
- In the main text it should be mentioned that CVD diamond is used.
- One weakness of the paper is that the authors do not comment on the reflection properties of the mirror under off-normal incidence. The demonstrated performance is only for normal incidence also simulations are not performed for different incidence angles. Mirrors are usually used under 45deg angle. The authors should comment this.
- I suggest introducing headings (for example "Results", "Discussion", ...) for the individual sections.

Reviewer #3 (Remarks to the Author):

The contribution submitted by H. Atikian and co-authors presents nanostructured surfaces in single crystal diamond that exhibit high reflectivity, which is of great practical use for optical components in high-power laser applications, in view of the high laser damage threshold that can be achieved. The research groups involved, are well known in their respective fields, and have demonstrated numerous nanostructured meta-surfaces for flat optics [e.g. cited ref. 40] and diamond nanostructures for photonics [e.g. cited ref. 58]. The contribution combines the concept of nanostructures for high reflectivity with the methods for nano-structuring of single crystal diamond, and optical characterization is performed on the fabricated component, including demonstration of high laser damage threshold. The underlying photonic concepts are well understood, and numerous publications have demonstrated high reflectivity with guided mode resonances such as high contrast gratings over the past decades [1]. Recent progress has also been shown in etching nanostructures in single crystal diamond [2], and over the past decade, micro-optics [3] and diffractive optical elements [4], high reflectivity gratings [5] for telecom lasers as well as a wealth of nanophotonic structures [6] such as for quantum nanophotonics [7], have been demonstrated in diamond ([3,4] in poly, [4,5,6,7] in single crystal diamond). The study is without doubt of great interest to the specialized community and has been performed with great care, including theoretical background, numerical simulation and modeling, nanofabrication, detailed experimental characterization, carefully redacted methods section, and critical discussion. However, in view of the previously demonstrated work in the field, the contribution does not appear to represent an advance in understanding likely to influence thinking in the field, and a discernible reason why the work deserves the visibility of publication in Nature Communications rather than the best of the specialist journals is not evident. In view of high technical quality and the interest to the specialized community, the publication is best to be considered for publication in a more specialized journal.

[1] G. Finco et al. "Guided-mode resonance on pedestal and half-buried high-contrast gratings for biosensing applications" Nanophotonics, vol. , no. , 2021, pp. 20210347.

<https://doi.org/10.1515/nanoph-2021-0347>

- [2] F. Lenzini et al., (2018), Diamond as a Platform for Integrated Quantum Photonics. *Adv. Quantum Technol.*, 1: 1800061. <https://doi.org/10.1002/qute.201800061>
- [3] M. Karlsson et al., "Diamond micro-optics," *IEEE/LEOS International Conference on Optical MEMS*, 2002, pp. 141-142, <https://doi.org/10.1109/OMEMS.2002.1031483>
- [4] T. V. Kononenko et al., "Fabrication of diamond diffractive optics for powerful CO₂ lasers via replication of laser microstructures on silicon template, *Diamond and Related Materials*", 101, 107656, 2020. <https://doi.org/10.1016/j.diamond.2019.107656>
- [5] G. Huszka et al. "Single crystal diamond gain mirrors for high performance vertical external cavity surface emitting lasers" *Diamond and Related Materials* 104, 107744, 2020. <https://doi.org/10.1016/j.diamond.2020.107744>
- [6] T. Schröder et al. "Quantum nanophotonics in diamond", *Journal of the Optical Society of America B* Vol. 33, Issue 4, pp. B65-B83 (2016) <https://doi.org/10.1364/JOSAB.33.000B65>
- [7] I. Aharonovich et al. (2014), "Diamond Nanophotonics", *Advanced Optical Materials*, 2: 911-928. <https://doi.org/10.1002/adom.201400189>

**** See Nature Research's author and referees' website at www.nature.com/authors for information about policies, services and author benefits.**

Response to the reviewers' comments

We provide a point-by-point response to the (italicized) reviewers' comments below using red text.

Reviewer #1 (Remarks to the Author):

The manuscript reports the design, fabrication and testing of a monolithic diamond high reflector that uses the resonant properties of an optimized nanostructure. Like moth eye anti-reflections structures, the aim is to make the optic from a single material, in this case one with extreme thermal and optical properties, to provide rapid heatsinking and also presumably to avoid differential expansion, thereby raising its power handling without damage. The work relies on the exquisite nanofabrication techniques of the team, which is world leading and has been used previously to develop a range of ground-breaking advances in diamond quantum optics, opto-mechanics and nanooptics. The mirror structures in this work are suitably backed up by a model of the em-field surface interaction. The main conclusions are that the fabricated mirror achieves a reflection coeff above 98% while withstanding 10 kW in 750 micron-diameter spot (equivalent to intensity 0.5 MW/cm² or linear damage threshold 0.13 MW/cm, to my calculations), a level of intensity that a sample off-the-shelf dielectric stack mirror was found to fail.

The results are noteworthy in that this is the first planar photonic crystal for high reflection (anti-reflection structures were demonstrated some years ago). On the point of the physics, the high reflectivity relies on a photonic resonance in the 'head' of the golf tee, which is an interesting outcome. The results may be of interest to those working in directed energy and materials processing applications. Since diamond transmits well in some areas of the infrared, including beyond 6 microns, the approach may be useful for long wavelength high power lasers such as CO₂ lasers. The authors suggest that it may also be helpful for CO₂ laser mirrors in the EUV lithography, although I believe that is a short-pulsed laser application so that more testing would have to be done to determine whether the mirrors are applicable to that application.

Thanks for the positive feedback. The CO₂ pump lasers in EUV lithography emit pulses with durations between 10 and 100 ns which constitute quasi-CW operation. Nonetheless, to your point, there has been investigations on using ps-duration pulses from a CO₂ pump. See, e.g., Oscar O Versolato 2019 Plasma Sources Sci. Technol. 28 083001.

To clarify this, we modified the sentence in the conclusion to:

"For example, applications in directed energy, manufacturing, or extreme ultraviolet lithography systems which often utilize quasi-CW high-power CO₂ lasers (pulse durations of 10-100 ns) will benefit."

We also add the mentioned reference, Oscar O Versolato 2019 Plasma Sources Sci. Technol. 28 083001, to the manuscript.

Note that we calculate the power density following what is described in the technical note found here: https://www.newport.com/medias/sys_master/images/h96/hb2/8797062889502/Calculating-Power-Density-Tech-Note-2.pdf

Thus, a 10 kW Gaussian-profiled laser beam over a 0.75 mm diameter spot produces a power density of 4.5 MW/cm².

A higher damage threshold is found compared to an off-the-shelf mirror Thorlabs broadband dielectric stack on glass, which hints (only) that the approach is promising for achieving higher damage thresholds than conventional mirrors. Most of the conclusions are well supported by the data – an exception is the aforementioned concern that the designs are not shown yet to be a benefit to the pulsed CO2 laser application. The conclusion sentence “We expect our work to stimulate development of a new class of optical components for CW high-power laser applications across the optical spectrum.” Is perhaps also not yet justified by the data – see point #6 below.

Our intention was to present this sentence as an outlook, but we now realize that it reads as a conclusion of our work. Thanks for pointing this out. We have modified this sentence to:

“Future work involves extending our approach to optical components for CW high-power lasers at other wavelengths”

The work is clearly explained (see also points below) and contains clear figures and useful supplementary material. I support publication in Nature Commun. with due consideration of the following points:

1) In the abstract, the sentence ‘these applications are unsuitable for mode-locked lasers’ is a little unclear – what applications are they referring to and is the intention to say that the applications are better served by CW lasers? This is a small point that could be addressed with some minor re-wording.

Thanks for mentioning this. We are not aiming to suggest that the applications are better served by CW lasers. To avoid confusion, and the fact that it is not critical to evaluating our work, we removed the sentence:

“Moreover, these applications are unsuitable for mode-locked lasers, which produce high intensity over only short (sub-ns) time intervals, and which induce optical damage by dielectric breakdown.”

2) Line 88: the novel features of the etching method are not clearly specified – is it only the rotation of the substrate that is new, or is the plasma design novel too?

The RIBAE etching method and plasma design has been used in a previous work, Atikian, H. A. et al. Freestanding nanostructures via reactive ion beam angled etching, APL Photonics 2, 051301 (2017), which we cited. Owing to the referee comment, we now realize we have overemphasized the novelty of our RIBAE etching method in the manuscript. Thus, we change “develop” to “use” and “employ” when referring to the RIBAE technique in the main text and Methods, respectively.

3) Fig 3a – it would be helpful to see a zoomed inset of the higher percentages, to make the 98% / 10nm claim more obvious.

We have added the zoomed inset as suggested.

4) The temperature images were taken with a FLIR camera. Please specify the model and state the uncertainty. Does the measurement rely on an emissivity value– if so, what value was used?

We double checked the camera specifications, finding that a Mikron M7600 with an accuracy of +/- 2 degrees was used to obtain the temperature images and not a FLIR camera. We regret our oversight on this. We set the emissivity of the Mikron camera to 0.97, a value that was recommended by Mikron for our measurements. From the experience of the Penn State Electro-Optics Center on measuring bare unpolished metals and glass/optical coatings with high power lasers, we also expect this emissivity parameter to give accurate results.

We have updated the Methods and the caption of Fig. 4 with information about the camera model, accuracy, and emissivity parameter.

5) 'The hot spot in the image suggests that 4% of laser power' – this is confusing – does the hotspot in the video images arise from emission from the diamond or the cooling plate behind? How does the hotspot suggest '4%'?

We apologize for the confusion from this sentence. Indeed, the thermal image and video does not allow determining the magnitude of laser power transmitted through the diamond onto the cooling plate behind. Since this mirror had 96% reflectivity, we aimed to say that up to 4% of the laser power incident on the mirror could have led to the heating observed by the thermal camera. We have changed the relevant sentence in the main text to:

"The hot spot in the images suggests that the laser power which is not reflected is, rather, transmitted through the mirror, heating the underlying water-cooled stage."

Annotation of the video point out the intersection of the beam with the diamond and the hotspot on the heatsink would be helpful.

We have updated the captions of the thermal videos (in Supplementary) to indicate the hotspots. Accordingly, we have added:

"The hot spot is directly below the number "1" label."

For the non-thermal video of the illuminated diamond mirror, we added the following clarification in its caption:

"The diamond is held between two copper clamps, with the etched section of the diamond (i.e. the mirror) appearing grey in color."

6) The diamond mirror is compared to that for a broadband HR mirror – this is not a high bar for high power mirror performance. A better choice would have been a narrowband mirror, with their fewer layers and much higher damage thresholds (twice as high for Thorlabs and 0.1 MW/cm is standard for IBS coatings). It would be helpful if the authors had data available for a mirror that is designed for high power. The study currently only shows that the microstructures mirrors work and provide reasonably high damage threshold. If an advantage over other coatings were to be demonstrated, a better experiment would be to compare their microstructured diamond mirror with a HR coating (eg., IBS) on a diamond substrate, in this way unambiguously showing the advantage of the structure over conventional

coatings (the choice of substrate is also likely to play a role). I suggest the authors comment on this possibility in their discussion.

We aimed to simply demonstrate a highly reflective monolithic diamond mirror that withstands high CW optical power, using a beam diameter that is representative of a practical laser system. Our results, we believe, stand on their own, even without a comparison. However, we felt compelled to give context to our work for the general reader of Nature Communications, hence the comparison with the Thorlabs mirror. Nevertheless, we certainly agree with the argument that there are higher thresholds to compare our mirror against, such as mirrors with ion-beam-sputtered or evaporated coatings.

Consider a manuscript by DeBell (citation below), which provided an overview of CW laser damage test results of reflectors made using IBS and evaporated coatings. They mention damage thresholds for infrared laser light ranging between 0.1 and 10 MW/cm² depending on the laser spot size (compare with 4.5 MW/cm², 10 kW into 750 μm spot diameter, as mentioned above, for our mirror) but without any important details such as overall laser power used, mirror reflectivity, composition, or thickness, and minimal details about fabrication. Consider also a continuous-wave laser damage test by Brown et al. (citation below). They used hafnia-silica and tantala-silica IBS mirrors, and even performed the test at the same facility (Penn State Electro-Optic Center) and wavelength (1070 nm) as our test. Powers (up to 17 kW), and spot sizes (1 mm diameter) were similar to those used in our test as well. Their work showed no damage with irradiances up to 3 MW/cm². However, any information regarding the reflectivity magnitude or spectrum, or fabrication process details of these mirrors were again not mentioned. Note that there has been work by Anokin et al. (citation below) on testing anti-reflection coated CVD diamond substrates (coating composition not revealed) for diamond windows. The work demonstrated that, using 1070 nm wavelength light, powers of a few hundred Watts on a 91 μm diameter spot damaged the window (compare with the 10 kW on a 750 μm diameter spot used in our work). We acknowledge that the anti-reflection coating would direct more light onto the sample, and hence could not constitute a direct comparison to our approach.

Clearly, due to the limited information that is revealed in the tests results published in literature (partially because this is likely proprietary), it is hard to directly compare the performance of our mirror to the competition. Such available information could also be highly valuable in, for example, allowing identification of the key challenges in coating development or motivating further improvements in monolithic designs, e.g. minimizing fabrication imperfections. Indeed, it is well known in the laser damage community that when trying to compare one technique to another, the varying protocols and approaches used to both characterizing an optic and performing laser-damage tests can lead to skewed results. Furthermore, the lack of a standard ISO testing protocol for CW laser damage hampers comparisons between optics. Note that Stolz and Negres (see “Ten-year summary of the Boulder Damage Symposium annual thin film laser damage competition” *Optical Engineering*, 57(12), 121910, 2018) have attempted to establish testing metrics for pulses laser damage tests with over 10 years of data. We also point out that, while performing tests using a thin film deposited on a diamond substrate would seem to be a better “apples-to-apples” comparison, without significant effort in optimizing the coating, it would be easy to skew the data in our favor, giving a misleading conclusion.

Nevertheless, we expect that our approach offers greater potential for the future development of high-power CW optics because it is (i) monolithic and (ii) single crystal, meaning there are few opportunities for light to be absorbed and be converted into heat and strain, and that is (iii) made of diamond, a material with superior thermal-mechanical and thermal-conductive properties.

In view of the referee comment, we have added the following sentence to the discussion:

“Further tests of the damage thresholds of the monolithic diamond mirror against custom high-power mirrors, e.g. those relying on ion-beam-sputtered or evaporated coatings on varied dielectrics, including diamond, would be valuable to determine the full extent of our mirror relative to the state of the art, see e.g., Refs [DeBell], [Brown et al], and [Anoikin et al].”

We have added the mentioned citations to the manuscript:

DeBell, Gary W. "Ion beam sputtered coatings for high fluence applications," Proc. SPIE 5991, Laser-Induced Damage in Optical Materials: 2005, 599116 (2006).

Andrew Brown, et al. "Continuous-wave laser damage and conditioning of particle contaminated optics," Appl. Opt. 54, 5216-5222 (2015).

Anoikin, Eugene, et al. "Diamond optical components for high-power and high-energy laser applications." Components and Packaging for Laser Systems. Vol. 9346. International Society for Optics and Photonics (2015).

This is an interesting paper that points towards new possibilities for high power mirror structures. I look forward to seeing the final article in publication.

Rich Mildren

Thanks for the careful evaluation of the manuscript and support for its publication.

Reviewer #2 (Remarks to the Author):

The paper is well written and the described subject is original with respect to the state-of-the-art. The demonstrated properties of the designed and manufactured diamond mirror with ion-etched nanostructure offer an interesting opportunity for high-power CW-applications and are relevant to the field of high-power laser optics. I suggest to publish the paper after the authors have made a few minor changes to their manuscript.

Thanks for the positive synopsis and suggestion to publish.

I have the following suggestions and comments for the authors:

- *In Fig 1D, I suggest putting the quantity's name "reflectivity" on the false color scale.*

We implemented the modification.

- In Fig 1E, I suggest putting the quantity's name "electric field amplitude" on the false color scale.

We implemented this modification as well.

- Line 82: "at vertical incidence" -> "at normal incidence"

Implemented this too.

- In Fig 3B, I suggest adding x- and y-axis and not just putting a scale bar into the diagram with the value only found in the captions. There is no label on the color bar. Also change "normalized optical power" to "normalized optical intensity".

Apologies for missing this. We have added the axes and labelled the color bar. We have also updated the rest of the figure and caption accordingly.

- Can the authors comment why they use single crystal diamond and not polycrystal diamond?

We use single crystal rather than polycrystal to avoid defects and domain walls which can absorb light and consequently produce heat that damages the diamond lattice. Of course, we could fabricate our design using polycrystal diamond, but this would produce a mirror with degraded performance compared to single crystal diamond.

- The state-of-the-art regarding diamond as a high-power material for optical components is not described well. There are for example works that demonstrate the use of single crystal diamond lenses (with dielectric coatings) sustaining kW CW laser power: C. Holly, M. Traub, D. Hoffmann, C. Widmann, D. Brink, C. Nebel, T. Gotthardt, M. Ceyhan Sözbir, and C. Wenzel, "Monocrystalline CVD-diamond optics for high-power laser applications," in *High-Power Laser Materials Processing: Lasers, Beam Delivery, Diagnostics, and Applications V*, Proc. SPIE, vol. 9741, 2016.

We regret missing this reference. We have added it to the manuscript in the following sentence:

"Consequently, diamond materials, in particular optics, can be used in a diverse range of applications and operating environments, see e.g., Refs. [Holly et al., Karlsson et al., Kononenko et al., Huszka et al., Aharonovich et al.] and references therein."

We have also added the following references, as suggested by Referee #3:

M. Karlsson et al., "Diamond micro-optics," IEEE/LEOS International Conference on Optical MEMs, 2002, pp. 141-142.

T. V. Kononenko et al., "Fabrication of diamond diffractive optics for powerful CO₂ lasers via replication of laser microstructures on silicon template, *Diamond and Related Materials*", 101, 107656, 2020.

G. Huszka et al. "Single crystal diamond gain mirrors for high performance vertical external cavity surface emitting lasers" *Diamond and Related Materials* 104, 107744, 2020.

I. Aharonovich et al. "Diamond Nanophotonics", *Advanced Optical Materials*, 2: 911-928 (2014).

- *Line 371: The "2" can be subscripted.*

We have fixed this and other places in the text where chemical formula subscripts were missed.

- *Line 410: What is the goodness of the fit?*

A χ^2 of 0.002 for a Gaussian distribution is obtained from the fit. We added this information to line 410.

- *Why is the laser focused onto the mirror for reflectivity measurements? This creates a spectrum of incidence angles around the normal incidence while with a collimated beam the angular spectrum is reduced.*

This was for reasons as discussed in the manuscript:

"This spot size represents a reasonable beam diameter that would be used in a practical laser system, corresponding to a Rayleigh length of roughly 1.66 m for a Gaussian beam at this wavelength. Smaller spot sizes can be used to increase the power density; however, we attempt to perform the LIDT tests with beam waists, and associated thermal loading, that would be present in a realistic optical system." Moreover, in these conditions, the beam is nearly collimated, thereby avoiding the range of incidence angles that you suggested.

- *Extended Data Fig 4: Put label on vertical axis.*

Thanks for catching that. We now say "Intensity (a.u.)".

- *In the main text it should be mentioned that CVD diamond is used.*

We modified the sentence in the main text to:

"To realize these complex 3-D structures across a wide area, we develop an unconventional, yet scalable, angled-etching nanofabrication technique for single-crystal chemical vapor deposition diamond, as illustrated in Fig. 2 and described in its caption."

- *One weakness of the paper is that the authors do not comment on the reflection properties of the mirror under off-normal incidence. The demonstrated performance is only for normal incidence also simulations are not performed for different incidence angles. Mirrors are usually used under 45deg angle. The authors should comment this.*

We designed a mirror to operate at normal incidence as to direct the maximum energy of the laser beam onto the mirror. Using a 45 degree angle would result in a lower surface temperature, and hence would not allow us to conclude the maximum impact intensity. Designing a mirror for a 45 degree angle is straightforward, as guided resonance structures can be designed for wide-angle impact. Nevertheless, we chose to operate at normal incidence to clearly demonstrate the most extreme operating conditions.

To clarify this point, we added the following sentence:

“Moreover, we perform tests at normal incidence, ensuring maximum energy is directed at the mirror.”

- I suggest introducing headings (for example “Results”, “Discussion”, ...) for the individual sections.

We have made the changes to the manuscript accordingly.

Reviewer #3 (Remarks to the Author):

The contribution submitted by H. Atikian and co-authors presents nanostructured surfaces in single crystal diamond that exhibit high reflectivity, which is of great practical use for optical components in high-power laser applications, in view of the high laser damage threshold that can be achieved. The research groups involved, are well known in their respective fields, and have demonstrated numerous nanostructured meta-surfaces for flat optics [e.g. cited ref. 40] and diamond nanostructures for photonics [e.g. cited ref. 58]. The contribution combines the concept of nanostructures for high reflectivity with the methods for nano-structuring of single crystal diamond, and optical characterization is performed on the fabricated component, including demonstration of high laser damage threshold. The underlying photonic concepts are well understood, and numerous publications have demonstrated high reflectivity with guided mode resonances such as high contrast gratings over the past decades [1]. Recent progress has also been shown in etching nanostructures in single crystal diamond [2], and over the past decade, micro-optics [3] and diffractive optical elements [4], high reflectivity gratings [5] for telecom lasers as well as a wealth of nanophotonic structures [6] such as for quantum nanophotonics [7], have been demonstrated in diamond ([3,4] in poly, [4,5,6,7] in single crystal diamond). The study is without doubt of great interest to the specialized community and has been performed with great care, including theoretical background, numerical simulation and modeling, nanofabrication, detailed experimental characterization, carefully redacted methods section, and critical discussion. However, in view of the previously demonstrated work in the field, the contribution does not appear to represent an advance in understanding likely to influence thinking in the field, and a discernible reason why the work deserves the visibility of publication in Nature Communications rather than the best of the specialist journals is not evident. In view of high technical quality and the interest to the specialized community, the publication is best to be considered for publication in a more specialized journal.

[1] G. Finco et al. "Guided-mode resonance on pedestal and half-buried high-contrast gratings for biosensing applications" *Nanophotonics*, vol. , no. , 2021, pp. 20210347.

<https://doi.org/10.1515/nanoph-2021-0347>

[2] F. Lenzini et al., (2018), *Diamond as a Platform for Integrated Quantum Photonics*. *Adv. Quantum Technol.*, 1: 1800061. <https://doi.org/10.1002/qute.201800061>

[3] M. Karlsson et al., "Diamond micro-optics," *IEEE/LEOS International Conference on Optical MEMs*, 2002, pp. 141-142, <https://doi.org/10.1109/OMEMS.2002.1031483>

[4] T. V. Kononenko et al., "Fabrication of diamond diffractive optics for powerful CO₂ lasers via replication of laser microstructures on silicon template, *Diamond and Related Materials*", 101, 107656, 2020. <https://doi.org/10.1016/j.diamond.2019.107656>

[5] G. Huszka et al. "Single crystal diamond gain mirrors for high performance vertical external cavity

surface emitting lasers” *Diamond and Related Materials* 104, 107744, 2020.

<https://doi.org/10.1016/j.diamond.2020.107744>

[6] T. Schröder et al. “Quantum nanophotonics in diamond”, *Journal of the Optical Society of America B* Vol. 33, Issue 4, pp. B65-B83 (2016) <https://doi.org/10.1364/JOSAB.33.000B65>

[7] I. Aharonovich et al. (2014), “Diamond Nanophotonics”, *Advanced Optical Materials*, 2: 911-928.

<https://doi.org/10.1002/adom.201400189>

Although we humbly accept the praise offered by the referee about our group and work, we disagree with their assessment that our manuscript is unsuitable for Nature Communications.

We remind the referee that, our work, demonstrating a monolithic diamond mirror for high-power continuous-wave lasers, has the potential to transform, and promote, the use of continuous-wave lasers. These lasers are used in a wide range of industries and applications, as pointed out in our manuscript.

Indeed, the impact and broad (real-world) utility of our work crucially relies on the definitive proof of high-power compatibility and high reflectivity that we provide—a result that was not obvious prior to our investigation for several reasons, e.g. material properties or fabrication imperfections and tolerances. Accordingly, our monolithic diamond mirror goes significantly beyond previous work on diamond metasurfaces and nanostructures, which is largely relevant to academic communities and not compatible with high power continuous-wave laser applications.

Specifically concerning the comment about publishing in Nature Communications compared to “the best of the specialist journals” and of our manuscript’s “high technical quality and the interest to the specialized community”, we want to remind the referee of the scope of Nature Communications, which reads “Papers published by the journal aim to represent important advances of significance to specialists within each field”. Given that the referee is familiar with such a scope, we find the referee’s comments somewhat contradictory to their negative assessment of publishing in Nature Communications.

We also thank the referee for providing these references. We have added Refs. [3-5, 7] above, which fall most closely within the scope of our work.

REVIEWERS' COMMENTS

Reviewer #1 (Remarks to the Author):

Of my 6 main comments, all have been comprehensively addressed.

I had also commented upon the applicability of the mirrors to the EUV lithography application, and that this should be removed as such pulses would be in a different regime. The authors suggest that the 10-100 nanosecond pulse durations can be considered quasi-cw. In the context here that relates to laser damage, quasi-cw implies that the damage threshold and damage mechanisms are in the same or similar regime as cw. This is not the case – the ns damage mechanism is generally regarded as being seeded via electrostrictive ejection of defects. In contrast, cw or quasi-cw damage seeding mechanisms are thermal in nature (eg., melt, vapourization, thermal delamination). Hence the applicability of the mirrors to the EUV application is speculative and not well supported. Unless evidence is presented in the paper damage threshold advantages in the nanosecond regime, I suggest that this application be removed or at least the background above be incorporated to provide readers with a reasonable assessment of how likely these mirrors will benefit that application. (It would be interesting to learn if the fast thermal response of the pillars influences the ns damage mechanism in this case!) Finally, from a terminology perspective, I suspect that the majority of readers wouldn't regard ns pulses as quasi-cw, except perhaps a few in the ultrafast community.

This point is not a core issue and therefore changes are very much at the discretion of the authors.

Reviewer #2 (Remarks to the Author):

The authors have responded to my review comments and made corresponding changes to their manuscript. The comments / questions were answered with care and the suggestions are now reflected in the updated version.

One style remark: The citation style is different in line 154. The authors use super-scripts in the rest of the text.

The work can be published after this minor correction has been made.

We provide a point-by-point response to each reviewer below in red.

Reviewer #1 (Remarks to the Author):

Of my 6 main comments, all have been comprehensively addressed.

Our response: Thank you again for the comments. The manuscript has been significantly improved because of them.

I had also commented upon the applicability of the mirrors to the EUV lithography application, and that this should be removed as such pulses would be in a different regime. The authors suggest that the 10-100 nanosecond pulse durations can be considered quasi-cw. In the context here that relates to laser damage, quasi-cw implies that the damage threshold and damage mechanisms are in the same or similar regime as cw. This is not the case – the ns damage mechanism is generally regarded as being seeded via electrostrictive ejection of defects. In contrast, cw or quasi-cw damage seeding mechanisms are thermal in nature (eg., melt, vapourization, thermal delamination). Hence the applicability of the mirrors to the EUV application is speculative and not well supported. Unless evidence is presented in the paper damage threshold advantages in the nanosecond regime, I suggest that this application be removed or at least the background above be incorporated to provide readers with a reasonable assessment of how likely these mirrors will benefit that application. (It would be interesting to learn if the fast thermal response of the pillars influences the ns damage mechanism in this case!) Finally, from a terminology perspective, I suspect that the majority of readers wouldn't regard ns pulses as quasi-cw, except perhaps a few in the ultrafast community. This point is not a core issue and therefore changes are very much at the discretion of the authors.

Our response: Thanks for the insight on the electrostrictive-based damage mechanism for ns-duration pulses. To avoid any confusion, we removed the mention of EUV applications in the Introduction and Discussion, along with associated references.

Reviewer #2 (Remarks to the Author):

The authors have responded to my review comments and made corresponding changes to their manuscript. The comments / questions were answered with care and the suggestions are now reflected in the updated version.

One style remark: The citation style is different in line 154. The authors use super-scripts in the rest of the text.

The work can be published after this minor correction has been made.

Our response: Thanks for the remarks. The manuscript has been greatly improved due to your comments. We have updated the citation style.